# Potential of Karrikins as Novel Plant Growth Regulators in Agriculture

**DOI:** 10.3390/plants9010043

**Published:** 2019-12-26

**Authors:** Michal Antala, Oksana Sytar, Anshu Rastogi, Marian Brestic

**Affiliations:** 1Department of Plant Physiology, Faculty of Agrobiology and Food Resources, Slovak University of Agriculture, 94976 Nitra, Slovakia; 2Department of Plant Biology, Educational and Scientific Center “Institute of Biology and Medicine”, Kiev National University of Taras Shevchenko, Volodymyrska, 64, 01033 Kyiv, Ukraine; 3Laboratory of Bioclimatology, Department of Ecology and Environment Protection, Poznan University of Life Sciences, Piątkowska 94, 60-649 Poznań, Poland; 4Department of Botany and Plant Physiology, Faculty of Agrobiology, Food and Natural Resources Czech University of Life Sciences, 16500 Prague, Czech Republic

**Keywords:** karrikins, seed germination, crops, *Arabidopsis*

## Abstract

Karrikins (KARs) have been identified as molecules derived from plant material smoke, which have the capacity to enhance seed germination for a wide range of plant species. However, KARs were observed to not only impact seed germination but also observed to influence several biological processes. The plants defected in the KARs signaling pathway were observed to grow differently with several morphological changes. The observation of KARs as a growth regulator in plants leads to the search for an endogenous KAR-like molecule. Due to its simple genomic structure, *Arabidopsis* (*Arabidopsis thaliana* L.) helps to understand the signaling mechanism of KARs and phenotypic responses caused by them. However, different species have a different phenotypic response to KARs treatment. Therefore, in the current work, updated information about the KARs effect is presented. Results of research on agricultural and horticultural crops are summarized and compared with the findings of *Arabidopsis* studies. In this article, we suggested that KARs may be more important in coping with modern problems than one could imagine.

## 1. Introduction

Agriculture of the twenty-first century must face new challenges, which require novel solutions [1]. The use of industrial fertilizers, pesticides, and new varieties boosted the green revolution in the last century [2]. This century is even more challenging due to its faster-changing environment than civilization has ever faced, which makes us seek modern means of food and material production [3,4]. Not only does agriculture meet new challenges, but climate change brings along more and more forest and grassland fires [5]. Understanding how these will change biomes all around the world and what mechanisms are hidden behind changes is therefore crucial.

From the time when the evidence of a germination cue created by burning plant material was reported the first time [6], many studies exploring its nature have been performed. We now understand that a group of butenolide compounds isolated from smoke, the first member of which was identified independently by two researchers’ teams [7,8], plays a major role in germination promotion. Further on, another five analogs were added to this group [9], and the group was named KARs according to the word of Australian aborigines for smoke “karrik” [10]. In addition to KARs, cyanohydrins were identified as a germination cue, which can stimulate the germination of some KAR-insensitive species [11].

Smoke was observed to stimulate the germination of many species from different families [10,12]. KARs are the major germination promoting compound found in smoke, and the KAR receptor is present in all phylogenetic taxa of plants, including mosses, liverworts, or green algae [13]. Therefore, the germination of smoke-responsive species is very likely enhanced by KARs. Indeed, many studies reported that KARs elevated germination of dicotyledonous, as monocotyledonous plants belonging to different plant life forms—Annuals, perennials, woody plants, and different importance for people—Weeds or agricultural and horticultural crops [14,15,16,17,18,19,20,21,22]. KARs are a promising new plant growth regulators [23,24], and yet unidentified endogenous molecule perceived through the same signaling pathway is a potential new phytohormone [25]. 

However, not only germination stimulators, but also germination inhibitors belonging to butenolides are present in smoke [26]. The study of individually derived smoke from 27 different plant species revealed differences in seed germination of *Themeda triandra* [27]. Quantification of KARs in smoke water using ultrahigh-performance liquid chromatography-tandem mass spectrometry revealed differences in both total and relative levels of KAR [28]. The smoke water prepared by different protocols needs to be dissolved in different ratios due to the variable content of germination stimulators and inhibitors [29]. Therefore, utilization of KARs rather than aerosol smoke or smoke-water is convenient for research and agriculture, despite the lower price of aerosol smoke and smoke-water compared to KARs [30]. 

Another possible means of KARs application to agriculturally utilized soils is indirectly through biochar, where KARs have been identified just recently [31]. Biochar has several other advantages in addition to KARs content. It improves nutrient sorption and water holding capacity of the soil. Carbon sequestration is another nonnegligible property of biochar [32]. Nevertheless, KARs content of biochar, like in case of smoke, is dependent on charred material and the used technology of pyrolysis. The individual plant species response relates to its sensitivity to KARs and inhibitory compounds within the biochar [31].

In the current review, up-to-date known effects of KARs on *Arabidopsis thaliana* in different stages of ontogenesis are summarized and compared with plants of agricultural and horticultural importance. 

## 2. Chemical Properties of Karrikins

The structure of KARs is very similar to phytohormones strigolactones (SLs). Butenolide ring of KARs and lactone D ring of SLs are closely related [33]. The KARs structure combines a six-membered pyran ring with a five-membered butenolide ring. The differences between the six known KARs are based on methyl substitutions (Figure 1). These KARs are described as KAR_1_ to KAR_6_ [34]. Only C, H, and O are present in the two-ring structures of KARs. The pure KARs have a melting point of 118–119 °C, and they are the substances of crystalline character. KARs can be quickly dissolved in organic solvents and mildly in water [35]. 

KARs relate to SLs because they share a specific type of lactone known as a butenolide fused to a pyran ring with the systematic name 3-methyl-2H-furo[2,3-c]pyran-2-one [35]. The plant-made signaling compounds SLs are synthesized from carotenoids. To date, the structure of at least 20 different naturally occurring SLs has been characterized [36]. In contrast, KAR molecules are not produced by the plant itself but are formed by heating or combustion of carbohydrates, such as cellulose [37]. Various SL analogs abbreviated as GR have been synthesized, of which GR24 is the most active and widely used in SL research [38,39]. 

Even though just six KARs so far showed physiological activity in plants, almost 50 analogs of KAR_1_ with different substitutions have been synthesized [40,41,42].

## 3. Perception of Karrikins by Plants

Although KARs were discovered 15 years ago, the exact mechanism of perception remains a mystery. However, it does not mean that we do not have any clue about signaling cascade, which begins by the sensing of KARs and ends by morphological and physiological responses. Genetic studies indicated that KARs are perceived by the KARRIKIN INSENSITIVE 2 (KAI2) receptor. KAI2 interacts with MORE AXILLARY GROWTH2 (MAX2), which leads to complex degrading SUPPRESSOR OF MAX2 1 (SMAX1) and SMAX1-LIKE2 (SMXL2). These reveal transcription factors from suppression and response to KARs occurs (Figure 2) [43,44,45,46].

The analyses of crystallography and ligand-binding experiments of KARs recognition by KAI2 revealed that KAR_1_ is capsulized through geometrically defined aromatic–aromatic interactions. KAR_1_ attachment induces the changes in KAI2 conformation at the active site entrance. The KAR_1_ ligand is located marginally at an active site distal from the catalytic triad (Ser95-His246-Asp217). Such location is consistent with the lack of detectable hydrolytic activity by purified KAI2 [45]. Just a single nucleotide mutation on KAI2 can considerably reduce the KAR-binding activity of KAI2. Mutation of codon 219 causing a change from alanine to valin alternates biochemical features of KAI2 and makes a plant severely or completely insensitive to KARs [47]. The KAI2 receptor protein is lost or degraded by a mechanism requiring a yet unidentified cell compartment, but it is independent of 26S proteasome or MAX2. Such loss is probably through enzymatic degradation, and it is rather the result of signaling than its cause [48]. 

F-box protein MAX2 has a shared role in KARs and SLs signaling, but plants can recognize SLs from KARs and react accordingly [43]. The SLs are synthesized from carotenoids and perceived via the α/β hydrolase DWARF 14 (D14) and the F-box protein MAX2 [33,49,50]. In contrast, KAR molecules are not produced by the plant itself but are formed by heating or combustion of carbohydrates, such as cellulose [37], and there is strong evidence that the MAX2-KAI2 protein complex might also recognize so far unknown plant-made KAR-like molecules [42]. Receptor KAI2 is important for cotyledon expansion, shortening petioles, and leaves to achieve wild type size, anthocyanins, and chlorophylls’ accumulation and enhanced expression of CHLOROPHYLL A/B BINDING PROTEIN 3 and CHALCONE SYNTHASE, which are light-responsive genes [51]. The proposed endogenous KAI2 ligand (KL) is not produced by the known SL biosynthesis pathway via carlactone [52]. 

Recent investigations of host perception in parasitic plants have demonstrated that SL recognition could evolve following gene duplication of KAI2. There are striking parallels in the signaling mechanisms of KARs, SLs, and other plant hormones, including auxins, jasmonates, and gibberellins (GAs) [24]. 

## 4. Effect of Karrikins on *Arabidopsis*

*Arabidopsis thaliana* (L.) has great value as a model plant with the sequenced genome [57] for studying all aspects of flowering plant life with a number of advantages [58]. It was an important finding that *Arabidopsis* is a KAR-sensitive plant, despite it not being a fire-following species [53].

The primary dormancy of *Arabidopsis* seeds can be overcome by KARs as it perceives KARs quickly and sensitively. KARs are an effective stimulator of seed germination, but they do not overcome the requirement for synthesis or perception of GAs. Amounts of GAs and abscisic acid (ABA) in seeds of *Arabidopsis* do not get changed in response to KARs during pre-germination [53]. KAR_2_ is the most effective KAR in germination stimulation and inhibition of hypocotyl elongation of *Arabidopsis* [53,54]. 

Inhibition of hypocotyl elongation and cotyledon expansion are light-dependent responses to KAR treatment. Under continuous red light, the KARs were observed to positively influence the accumulation of chlorophyll a and b in *Arabidopsis thaliana* [53]. KARs alone regulate germination and hypocotyl elongation of plants, whereas KARs together with SLs help in the regulation of leaf morphology in *Arabidopsis*. SLs repress branching and lower auxin transport [55]. 

Some of the root architecture features, which had been previously credited to SLs are actually regulated by KARs or by the interaction of SLs and KARs. KARs are responsible for hair root development, the direction of root growth, root diameter, and root waving. KARs and SLs together influence the density of lateral roots [56]. Previous confusion in the role of SLs was caused by the use of GR24 as a racemic mixture like an SL analog to study changes in plant development. This mixture is at the same time a potent activator of the SL signaling pathway due to the presence of natural stereoisomer GR24^5DS^ and the KAR signaling pathway by the non-natural stereoisomer GR24^ent-5DS^ [48,59].

## 5. Effects of Karrikins on the Crops’ Growth and Development 

Experiments of the KAR treatment effect has been done not only with model plants, but also with several crops as presented in Table 1. These studies are more valuable from a practical point of view as they provide cues about the advantages of KAR treatment for sustainable food production. 

KAR_2_ stimulates germination of *Arabidopsis* seeds under favorable conditions, but it can inhibit germination in the presence of osmolytes or at elevated temperature. KAI2 signaling may inhibit germination under unfavorable conditions as protection against abiotic stress [60]. However, germination and seedling growth of tef, an African cereal crop, under high temperature, and low osmotic potential were observed to be enhanced by KAR_1_ treatment [61]. The enhanced germination and improved tomato seedling development in temperature extremes connected with KAR_1_ utilization were also reported [62]. These facts show that the reactions of a model plant and crops can be different.

The level of ABA in imbibed seeds of *Arabidopsis* was not affected by KARs treatment [53]. That was not a case of *Avena fatua* kernels treated by KAR_1_, which showed a one-third decrease in the level of ABA after 16 hours of imbibition. A similar result was recorded for GA_3_ treatment. The promotion of germination in *Avena fatua* can be related to an increase in reactive oxygen species concentration, which may be a result of lower catalase and superoxide dismutase activity in the aleurone layer [63]. However, the endosperm of maize and cotyledons of bean showed higher antioxidation activity from the third day on, although antioxidant enzymes activity of roots, mesocotyl, and coleoptile of maize or embryo and shoot of the bean was either without change or lower. The improved seedling growth may be due to the movement of starch from storage parts of seeds to growing parts, and the increased activity of amylase in roots and aboveground parts [64]. Another study with different results than above-presented reports delayed germination of soybean after KAR_2_ treatment through enhancement of ABA biosynthesis and GA biosynthesis impairment [65]. This all shows the need for a study using one protocol to examine changes in germinating seeds of different species. In the absence of such studies, it is impossible to draw conclusions about the effect of KARs on biochemical changes during seed imbibition and germination of various species.

No significant influence of KARs has been reported on the primary root length of *Arabidopsis* [56], whereas a positive effect of KAR_1_ treatment on rice, tomato, okra, bean, maize, and carrot root was reported (Table 1). Not only the root length enhancement of rice was observed, but also the increased number of lateral roots was found [66]. This is the opposite of effect on *Arabidopsis*, where KARs repress lateral root development [56]. Thus, the effect of KARs on the root architecture of monocotyledonous and dicotyledonous plants may differ significantly. 

DWARF14LIKE, which is an *Arabidopsis* KAI2 analog in rice, is necessary for the initiation of colonization events by arbuscular mycorrhizal fungi, but KAR_2_ was not effective in colonization enhancement of wild-type roots by arbuscular mycorrhizal fungi [67]. Whether other KARs play some role in plant-fungi symbiosis or what another signal is perceived by KAI2 is for now unclear. 

*Arabidopsis* seedlings react more sensitively to light after the treatment by KARs, which results in shorter hypocotyl [54]. Interestingly, the majority of studied crops reacted by increased seedling height (Table 1). This seemingly opposite reaction can be explained as a response of seedlings under KAR treatment by the most convenient growth [54]. It is known that KARs are involved in the regulation of auxins biosynthesis [49,68,69]. Therefore, variability in growth may be caused by the different effects of KARs on auxins level in plants of different species.

Both *kai2* and *max2 Arabidopsis* mutants exhibit drought sensitivity. *Max2* and *kai2* mutants have larger stomatal aperture due to ABA-hyposensitivity, and both mutants also have a thinner cuticle. These result in higher water loss during dry periods. The rate of chlorophyll leakage in *max2* and *kai2* was observed to be higher than in wild type plants, suggesting that the evaporation through the cuticle of mutants is faster [70,71]. KAR_1_ improved the seedling performance of tomato and tef grown in lowered osmotic potential conditions [61,72]. These indicate the potential of KARs treatment for mitigation of drought stress effect on crops.

KARs stimulate chlorophyll concentration in *Arabidopsis*, tef, and carrot [21,54,73]. KARs not only influence the chlorophyll content, but also enhance net photosynthesis rate, probably as a result of increased stomatal conductance and higher intercellular CO_2_ concentration, which was found in KAR_1_ treated carrot plants [73]. However, foliar application of KAR_1_ on amaranth caused a reduction in chlorophylls content [74]. The mechanism behind the KARs influence on chlorophyll concentration and photosynthesis is, for now, unknown, but the method of application may be decisive. 

KAR signaling can also influence secondary metabolism. *Kai2* mutant of *Arabidopsis* has lower anthocyanin content as a result of transcription misregulation of the anthocyanin biosynthesis pathway [71]. Ascorbic acid and β-carotene content were increased in carrot roots grown from KAR_1_ primed seeds [73]. The content of tashinone I, pharmacologically active terpenoid, was significantly increased in hairy roots of *Salvia miltiorrhiza* by a signaling pathway involving nitric oxide and jasmonic acid [75].

Even though KAR_1_ improved plant height, weight, stem thickness, and the number of leaves of tomato, it did not increase the yield of fruits. However, fruits were observed to appear earlier on KAR_1_ treated plants than on the control plants, which can be advantageous for seasonal growers [76]. Similarly, grain yield of tef was not significantly improved, but stem thickness and plant height increased, which indicates the potential of higher hay yield interesting for animal farms [21]. Experiments with carrots indicate the considerable potential of KAR_1_ utilization for root yield quantity and quality enhancement. The carrot roots grown from KAR_1_ presoaked seeds were bigger, heavier, and contained more pigments than control plants [73]. 

KAR_1_ was tested for genotoxicity and mutagenicity on *Salmonella typhimurium* [77], in *Vicia faba* and *Persea Americana* metabolic activated Ames assay [78] and in juice from KAR_1_ treated onion by Ames assay [79]. The results of all tests do not show any genotoxicity nor mutagenicity. Therefore, KAR_1_ can be considered as safe for use in agriculture and horticulture. 

Utilization of KARs in dose 2–20 g ha^−1^ as weed control measure was proposed for agriculture [16]. Such use of KARs seems to be highly improbable as the cost of KARs would have to decrease thousands fold to reach an affordable level, and, even then, economic benefit for farmers would be questionable. More likely, KARs can be used as a priming agent for seeds of agricultural and horticultural crops in order to enhance germination and early seedling growth to establish a steady field under conditions of climate change. Priming of seeds is an efficient mean of application, and the positive effect of KARs on the vigor of plants grown from primed seeds endures for at least three months [72]. However, more studies are needed, which should be performed not only in the laboratory but mainly in field conditions, before agricultural practice accept such utilization as beneficial.

## 6. Conclusions

KARs are relatively simple molecules affecting several physiological and morphological features of different species. Their structure and signaling pathway are like plant hormones SLs. Finding that *Arabidopsis* is one of the KAR-responsive species enabled to study signaling cascade of KAR perception. Analysis of mutants shows that receptor KAI2 in complex with F-box protein MAX2 can degrade repressors SMAX1 and SMXL2, which release the number of genes from repression. That stimulates germination and cause morphological responses of aboveground and belowground organs. KARs can also stimulate the germination of several crops under optimal and suboptimal conditions. Responses of the model plant, *Arabidopsis*, and agricultural and horticultural crops are not always the same. Therefore, more studies on crops, mainly in field conditions, are needed to discover possible benefits of KARs use in the challenged nowadays agriculture.

## Figures and Tables

**Figure 1 plants-09-00043-f001:**
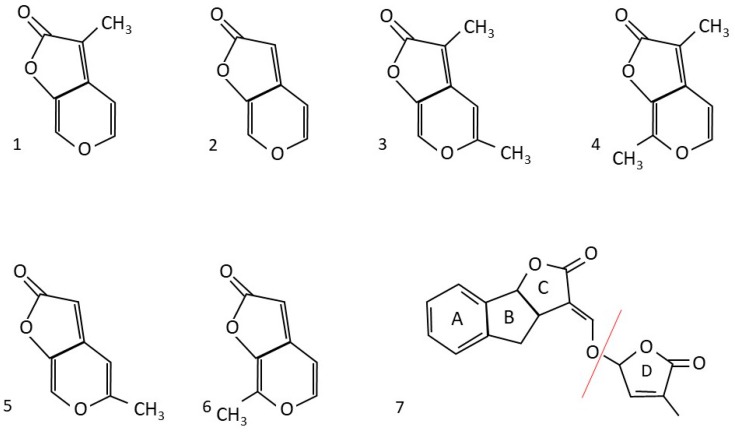
The known chemical structures of karrikin family representatives and strigolactone analog GR24. 1. KAR_1_ 2. KAR_2_. 3. KAR_3_, 4 KAR_4_, 5. KAR_5_, 6. KAR_6,_ 7. Strigolactone analog GR-24, the red line separates a lactone D ring, which is similar to the KARs butenolide ring.

**Figure 2 plants-09-00043-f002:**
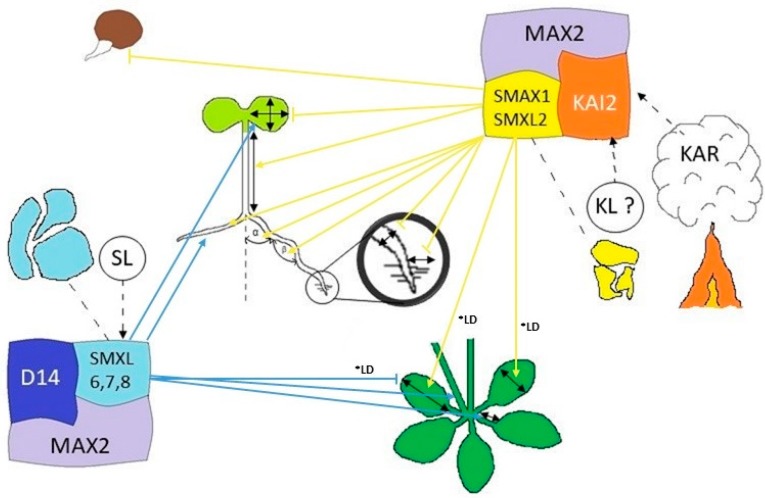
Model of signaling and effect of karrikins (KARs) and strigolactones (SLs) on *Arabidopsis* thaliana in different stages of ontogenesis. KARs produced by the burning of plant material and yet unidentified KAI2 ligand (KL) are perceived by the KAI2 receptor, which by interaction with F-box protein MAX2, causes degradation of SMAX1 and SMXL2. SMAX1 represses seed germination, SMAX1 and/or SMXL2 repress cotyledon expansion, root straightness, root width, and root hair development, and promote lateral root development and root skewness and hypocotyl elongation by reduction of seedling light sensitivity; SMAX1 promotes expansion of rosette leaves blade under long-day conditions (*****LD), SLs are perceived by receptor protein D14, which interacts with MAX2 and causes degradation of SMXL6,7,8. SMXL6,7,8 promotes cotyledon expansion, branching, and lateral root development; SMXL6,7,8 represses petiole and leaf blade elongation under long-day conditions (*****LD) [46,53,54,55,56].

**Table 1 plants-09-00043-t001:** Effects of karrikins (KARs) on the agricultural and horticultural crops growth, development and photosynthetic properties.

Plant	Conc. [M]	Means of Application	Examined Features	Effect of KAR	Ref.
Rice (*Oryza sativa* L.)	10^−10^–10^−8^	grown in Petri dishes with KAR_1_ solution	seedling weight, vigor index	+	[66]
			root and shoot length, no. of lateral roots	+	
Tomato (*Lycopersicon esculentum* Mill.)	10^−7^	grown in Petri dishes with KAR_1_ solution	germination	0	[80]
			% of abnormal seedlings	−	
			vigour index, seedling weight	+	
			hypocotyl and radicle length	+	
			weight of 10 embryonic axis	+	
			weight of 10 cotyledons	−	
Tomato (*Lycopersicon esculentum* Mill.), Okra (*Abelmoschus esculentus *L.), Bean (*Phaseolus vulgaris* L.) and Maize (*Zea mays *L.)	10^−7^	tomato, okra, bean and maize for germination experiment were grown in Petri dishes with KAR_1_ solution, maize kernels for growth experiment were presoaked in KAR_1_ solution for 1 h	**germination experiment:**		[20]
			root and shoot length	+	
			seedling weight of tomato. okra and maize	+	
			seedling weight of bean	0	
			vigor index	+	
			**growth experiment:**		
			fresh and dry weight of root	+	
			fresh and dry weight of shoot	+	
			no. of leaves, plant height	+	
			% of plant survival	+	
Tomato (*Lycopersicon esculentum* Mill.)	10^−7^	grown in Petri dishes with KAR_1_ solution in different temperatures	germination (t = 15, 20, 25, 30, 35 °C)	0	[62]
			germination (t = 10, 40 °C)	+	
			vigor index (all temperatures)	+	
			seedling weight (t = 20, 30, 35 °C)	0	
			seedling weight (t = 10, 15, 25, 40 °C)	+	
			root: shoot ratio (t =15, 20, 25, 30 °C)	0	
			root: shoot ratio (t = 10, 35, 40 °C)	+	
Tomato (*Lycopersicon esculentum* Mill.), Okra (*Abelmoschus esculentus *L.)	10^−7^	spraying by KAR_1_ solution to the point of runoff in four days intervals from eight day after seed sowing	shoot and root length	0	[81]
			shoot fresh and dry weight of okra	0	
			root dry weight of okra	−	
			shoot and root fresh weight of tomato	+	
			shoot and root dry weight of tomato	0	
			no. of leaves and total leaf area of tomato	+	
			no. of leaves and total leaf area of okra	0	
			stem thickness	0	
			seedling vigour and absolute growth rate	0	
Tomato (*Lycopersicon esculentum* Mill.)	10^−7^	seeds were primed in KAR_1_ solution for 24 h, blotted dry and grown in different temperatures or salt concentrations or osmotic potentials,	vigor index (salt concentration = 0, 100, 125, 150 mM)	+	[72]
			vigour index (Ψ_S_ = 0, −0.05, −0.15, −0.30, −0.49 MPa)	+	
			vigour index (t = 10, 15, 20, 25, 30, 35 °C)	+	
Tef (*Eragostis tef* Zucc.)	10^−8^	imbibed or grown in Petri dishes with KAR_1_ solution in different temperatures or osmotic potentials	imbibition (Ψ_S _= 0, −0.5 MPa)	0	[61]
			imbibition (Ψ_S _= −0.15, −0.30, −0.49 MPa)	+	
			germination	0	
			seedling length (t = 20 °C)	0	
			seedling length (t = 25, 30, 35, 40, 30/15 °C)	+	
			seedling length (Ψ_S _= 0, −0.5, −0.15, −0.30 MPa)	0	
			seedling length (Ψ_S _= −0.49 MPa)	+	
Tomato (*Lycopersicon esculentum* Mill.)	10^−9^	irrigation by KAR_1_ solution twice a week	plant height, plant weight	+	[76]
			no. of leaves, stem thickness	+	
			fruit appearance (days)	−	
			no. of fruits, fruit weight, fruit diameter	0	
			harvest index	+	
			ascorbic acid, β-carotene and lycopene content	0	
Onion (*Allium cepa *L.)	10^−10^	grown in pots drenched by KAR_1_ solution twice a week	no. of leaves, leaf length	+	[79]
			fresh and dry leaf weight	+	
			fresh bulb diameter	+	
			fresh bulb diameter	0	
			absolute growth	+	
			harvest index	0	
			genotoxicity and mutagenicity	0	
Pepper (*Capsicum annuum* L.), Salvia (*Salvia *sp.)	10^−7^	grown in pots irrigated by KAR_1_ solution	seedling emergence of pepper	+	[82]
			seedling emergence of salvia	0	
			seedling fresh and dry weight	+	
			mean emergence time (days)	−	
			catalase activity of pepper	0	
			catalase activity of salvia	+	
Tef (*Eragostis tef* Zucc.)	10^−8^	grown in pots drenched by KAR_1_ solution just once	leaf area, no. of tillers	0	[21]
			plant height, stem thickness	+	
			dry weight, grain yield	0	
			chlorophylls a and b	+	
			carotenoids	−	
Pepper (*Capsicum annuum* L.)	10^−7^	seeds presoaked in KAR_1_ solution for 40 h	germination, seedling emergence	+	[22]
			seedling fresh and dry weight	+	
			catalase activity	−	
			superoxid dismutase activity	+	
			ascorbate peroxidase activity	+	
Amaranth (*Amaranthus hybridus *L.)	10^−6^	grown in pots drenched by KAR_1_ solution once a week or foliar application or combination of drenching and foliar application	**drenching: no. of leaves and roots**	0	[74]
			shoot length	+	
			root length, stem thickness, leaf area	0	
			shoot fresh and dry weight	+	
			root fresh weight	0	
			root dry weight	−	
			**foliar: no. of leaves**	−	
			no. of roots, shoot and root length	0	
			stem thickness, leaf area	0	
			shoot fresh and dry weight	−	
			root fresh weight	0	
			root dry weight	−	
			chlorophylls a and b	−	
			carotenoids, protein content	+	
			carbohydrates content	−	
			**drenching + foliar: no. of leaves and roots**	0	
			shoot and root length, stem thickness, leaf area	0	
			root fresh and dry weight, shoot fresh weight	+	
			shoot dry weight	0	
Carrot (*Daucus carota *L.)	10^−10^–10^−7^	seeds presoaked in KAR_1_ solution for 12 h	germination, plant height	+	[73]
			leaf area, no. of leaves	+	
			length, diameter, fresh and dry weight of root	+	
			chlorophyll fluorescence (Fv/Fm)	+	
			net photosynthetic rate (P_N_)	+	
			stomatal conductance (gs)	+	
			intercellular CO_2_ concentration (Ci)	+	
			total chlorophyll content, carotenoids	+	
			β-carotene and vitamin C content of root	+	

Conc. means concentration, Ref. are references, Effect of karrikin (KAR): + means increase, − decrease and 0 no significant change of examined feature, KAR_1_ is karrikin_1_, KAR_2_ is karrikin_2_, t means temperature and Ψ_S_ means osmotic potential.

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
