# Peer review of "Potential of Karrikins as Novel Plant Growth Regulators in Agriculture"

_plants, 2019, doi:10.3390/plants9010043_

Round 1

Reviewer 1 Report

The subject of the manuscript is consistent with the scope of the Journal and is original and still not fully investigated; therefore this review could be a stimulation  for future researchs.

However, considering that the main field of this journal is the research of fundamental and applied fields of plant science, I think that section 5 “Effects of karrikins on the crops’ growth and development” should be more extensive. In fact, table 1 is a good summary of the main information obtained by previous studies, but they should be better discussed in the text.

The paper is well organized regarding the division into sections; however i have noted some mistakes and I suggest a moderate check.

In this paper there are several punctuation errors, for example in section 5 “Effects of karrikins on the crops’ growth and development” line 172, 173 and 214 at the end of sentence the fullstop is missing. Check the whole paper, please.

Some mistakes:

Section 5. “Effects of karrikins on the crops’ growth and development”

Line 222:g.ha-1” correct to g ha-1.

Table 1:

In this table, the title of second column is written in abbreviation “conc.” , please write it in full or explain it in a footnote; Enlarge the second column for allowing to write the values of concentrations in a single line (see first line –rice and last line -carrot) Some lines of the columns “Examined features” and “Effect of KAR” are not aligned (see line “Tomato” reference 78; “Tef” reference 61 and “Tomato” reference 79 and “Amaranth” reference 69. Please correct them.

Please, be sure that all the references cited in the manuscript are also included in the reference list and vice versa.

I recommend this publication with major corrections

Author Response

Dear Reviewer,

Thank you very much for the your useful comments. We tried to answer line to line to all your suggestions. Please see answers below.

Comments and Suggestions for Authors

The subject of the manuscript is consistent with the scope of the Journal and is original and still not fully investigated; therefore this review could be a stimulation for future researchs.

However, considering that the main field of this journal is the research of fundamental and applied fields of plant science, I think that section 5 “Effects of karrikins on the crops’ growth and development” should be more extensive. In fact, table 1 is a good summary of the main information obtained by previous studies, but they should be better discussed in the text.

Thank you, reviewer, for the comment and suggestions. We extended discussion related to content of table 1 and added a new paragraph about karrikins’ effect on yield of crops. The added changes are in red color.

The paper is well organized regarding the division into sections; however i have noted some mistakes and I suggest a moderate check.

In this paper there are several punctuation errors, for example in section 5 “Effects of karrikins on the crops’ growth and development” line 172, 173 and 214 at the end of sentence the fullstop is missing. Check the whole paper, please.

Thank you, for very accurate reading of our manuscript. Full stops were added and the whole paper was checked and updated. The changes are in red color.  

Some mistakes:

Section 5. “Effects of karrikins on the crops’ growth and development”

Line 222: “g.ha-1” correct to g ha-1.

Thank you that you noticed this mistake. We adjusted the unit according to your suggestion and SI.  

Table 1:

In this table, the title of second column is written in abbreviation “conc.” , please write it in full or explain it in a footnote; Enlarge the second column for allowing to write the values of concentrations in a single line (see first line –rice and last line -carrot) Some lines of the columns “Examined features” and “Effect of KAR” are not aligned (see line “Tomato” reference 78; “Tef” reference 61 and “Tomato” reference 79 and “Amaranth” reference 69. Please correct them.

Thank you for the correct comment which helped to improve the quality of the manuscript. Table 1 was formatted regarding the comments. The mistakes were removed, and the text was updated.

Please, be sure that all the references cited in the manuscript are also included in the reference list and vice versa. 

Thank you for the suggestions. The list of references was updated and checked for the presence of each reference in the text.

Reviewer 2 Report

I carefully read the ms entitled “Karrikins as novel plant growth regulators in applied 2 agriculture”. The ms summarizes the main findings and knowledge about KARs, focusing on their chemical properties, signalling mechanisms for their perceptions and effects in the model plant A. thaliana and crops. The organization of the ms is logical. However, despite the topic of the ms is very interesting, several parts require extensive improvement. English must be also absolutely revised throughout the ms. Because of poor English and styling mistakes, reading some parts of the text has been very hard to me, especially section 3 (Perception of karrikins by plants). The middle part of the abstract about Arabidopsis (from line 20 to 25) needs to be rewritten.

Lines 61-64: from this sentence I do not understand why using KARs may be more convenient. Which is the higher benefit for users? What the authors mean with "apertures"?

Based on the title, I expected more description about agricultural uses of KARs, which are only mentioned or briefly described. Therefore, I would suggest changing the title to make is more coherent to the text in the ms.

Author Response

Dear Reviewer,

Thank you very much for the your useful comments and suggestions. We tried to work on the MS text regarding your suggestions. The responses on your comments please see below.

Comments and Suggestions for Authors

I carefully read the ms entitled “Karrikins as novel plant growth regulators in applied 2 agriculture”. The ms summarizes the main findings and knowledge about KARs, focusing on their chemical properties, signalling mechanisms for their perceptions and effects in the model plant A. thaliana and crops. The organization of the ms is logical. However, despite the topic of the ms is very interesting, several parts require extensive improvement. English must be also absolutely revised throughout the ms. Because of poor English and styling mistakes, reading some parts of the text has been very hard to me, especially section 3 (Perception of karrikins by plants). The middle part of the abstract about Arabidopsis (from line 20 to 25) needs to be rewritten.

Thank you, reviewer, for the comment and suggestions.  The English language was revised by a premium account at grammarly.com.  More than 80 mistakes were repaired with the help of this application and stylization of many sentences was improved. We must admit that part “Perception of karrikins by plants”, mainly its second paragraph, was written in a very complicated and unclear way. Therefore, it was rewritten more simply and clearly. The middle part of the abstract about Arabidopsis was rewritten as well. The changes in red color.

Lines 61-64: from this sentence I do not understand why using KARs may be more convenient. Which is the higher benefit for users? What the authors mean with "apertures"?

Dear reviewer, thank you for these questions. From the mentioned lines it is maybe not clear, why the use of karrikins may be more convenient. However, we meant the whole paragraph (in revised manuscript lines 54-63) as justification for our thesis, that KARs may be more convenient for research and agriculture. We meant all equipment used for smoke water production by “apertures”. As this can be unclear, we substituted “apertures” by “protocols”, which does not change meaning, but can be clearer for readers.

Based on the title, I expected more description about agricultural uses of KARs, which are only mentioned or briefly described. Therefore, I would suggest changing the title to make is more coherent to the text in the ms.

Thank you, reviewer, for the comment and suggestion regarding the title of the manuscript.  We could not describe agricultural uses of KARs very extensively, because they are still not used practically. Our review should point out that some research in this field has been done and there is potential for use of KARs in agriculture. Therefore, we changed the title of manuscript to “Potential of karrikins as novel plant growth regulators in agriculture”.

Reviewer 3 Report

Generally, the manuscript is well written and is important, especially because of its practical significance. This review addresses current research questions concerning karrikins and gives some new insights into this compounds as novel plant growth regulators in applied agriculture. The manuscript brings an important perspective for the understanding of how plants can cope with butenolide compounds. I do believe that this paper is relevant and that also fits well to Plants.

Only one suggestion:

Conclusions need to also include the limitations of such study and further experiments to answer important questions resulting from performed so far. Are there any implications for future research?

Author Response

Dear Reviewer,

Thank you very much for the your suggestions and comments which are significantly improved presented MS. Please, find below responses for your comments.

Comments and Suggestions for Authors

Generally, the manuscript is well written and is important, especially because of its practical significance. This review addresses current research questions concerning karrikins and gives some new insights into this compounds as novel plant growth regulators in applied agriculture. The manuscript brings an important perspective for the understanding of how plants can cope with butenolide compounds. I do believe that this paper is relevant and that also fits well to Plants.

Only one suggestion:

Conclusions need to also include the limitations of such study and further experiments to answer important questions resulting from performed so far. Are there any implications for future research?

Dear Reviewer, thank you for the suggestion. The conclusion was updated regarding comments.

Reviewer 4 Report

The manuscript is interesting for agricultural and  horticultural. The aim  in the current review was effects of KARs on Arabidopsis thaliana in different stages of ontogenesis are summarized and discussed with effects on plants of agricultural and  horticultural importance..The subject of the manuscript is consistent with the scope of the Journal. The paper probably adds new and original values to the gricultural and  horticultural.

The present work is prepared in the usual manner for scientific work, both the division into chapters, the selection of references and collected results in the form of tables and figures. Given the importance of the problem studied, the presentation of the results and their interpretation, and the same content and form of the work, which I think is very well written proposes to adopt the manuscript for publication after minor revision.

Please, be sure that all the references cited in the manuscript are also included in the reference list and vice versa with matching spellings and dates.

Author Response

Dear Reviewer,

Thank you for the positive input in presented MS. The response to your suggestion see below.

The manuscript is interesting for agricultural and  horticultural. The aim  in the current review was effects of KARs on Arabidopsis thaliana in different stages of ontogenesis are summarized and discussed with effects on plants of agricultural and  horticultural importance. The subject of the manuscript is consistent with the scope of the Journal. The paper probably adds new and original values to the gricultural and  horticultural.

The present work is prepared in the usual manner for scientific work, both the division into chapters, the selection of references and collected results in the form of tables and figures. Given the importance of the problem studied, the presentation of the results and their interpretation, and the same content and form of the work, which I think is very well written proposes to adopt the manuscript for publication after minor revision.

Please, be sure that all the references cited in the manuscript are also included in the reference list and vice versa with matching spellings and dates.

Dear Reviewer, thank you for the suggestion. The list of references was checked regarding presence of each reference in the text.

Round 2

Reviewer 1 Report

Dear Authors,

I appreciate your efforts for improving the paper, if also I think that it could still be improved (especially the section 5). You have been corrected some mistakes (fullstops, table 1, etc), but I noted others:

- what is A219V at line 121??

-  correct the numbering of "conclusions", it is 6. and no 5.

Please, check again the references.

Finally I suggest a further english revision.

Author Response

Dear Reviewer,

Thank you very much for the serious checking of resubmitted version.

We worked on section 5 and removed not correct sentences. The English revision was done. The English spelling in the text was checked and text was updated.

The numbering of "conclusions" was corrected. The References were checked. The all changes in a red colour.

For the your comments: 

what is A219V at line 121??

We made explanation: Mutation of codon 219 causing change from alanin to valin alternates biochemical features of KAI2 and makes a plant severely or completely insensitive to KARs.

Thank you for the your valuable suggestions.

Best,

Reviewer 2 Report

I have read the revised manuscript and I really appreciate the efforts made by the authors to address all the concerns and the extensive revision they made. The quality of the manuscript has been greatly improved and I think it can be now suitable for publication. 

Author Response

Dear Reviewer,

Thank you very much for the positive feedback for the updated manuscript. We checked MS regarding English minor spell check and updated version was resubmitted. 

Best,
